# Synthesis and Bioactivity of Thiosemicarbazones Containing Adamantane Skeletons

**DOI:** 10.3390/molecules25020324

**Published:** 2020-01-13

**Authors:** Van Hien Pham, Thi Phuong Dung Phan, Dinh Chau Phan, Binh Duong Vu

**Affiliations:** 1Drug R&D Center, Vietnam Military Medical University. No.160, Phung Hung Street., Phuc La ward, Ha Dong District, Hanoi 100000, Vietnam; phamvanhien181288@gmail.com; 2Department of Pharmaceutical Chemistry, Hanoi University of Pharmacy. No. 15, Le Thanh Tong Street, Hoan Kiem District, Hanoi 100000, Vietnam; pdungdhd@gmail.com; 3Hanoi University of Science and Technology. No.1, Dai Co Viet Street., Bach Khoa Ward, Hai Ba Trung District, Hanoi 100000, Vietnam

**Keywords:** adamantane derivatives, thiosemicarbazone, antimicrobial, cytotoxicity activity

## Abstract

Reaction of 4-(1-adamantyl)-3-thiosemicarbazide (**1**) with numerous substituted acetophenones and benzaldehydes yielded the corresponding thiosemicarbazones containing adamantane skeletons. The synthesized compounds were evaluated for their in vitro activities against some Gram-positive and Gram-negative bacteria, and the fungus *Candida albicans*, and cytotoxicity against four cancer cell lines (Hep3B, HeLa, A549, and MCF-7). All of them showed good antifungal activity against *Candida albicans*. Compounds **2c**, **2d**, **2g**, **2j** and **3a**, **3e**, **3g** displayed significant inhibitory activity against *Enterococcus faecalis*. Compounds **2a**, **2e**, **2h**, **2k** and **3j** had moderate inhibitory potency against *Staphylococcus aureus*. Compounds **2a**, **2e** and **2g** found so good inhibitory effect on *Bacillus cereus*. Compounds **2d** and **2h**, which contain (*ortho*) hydroxyl groups on the phenyl ring, were shown to be good candidates as potential agents for killing the tested cancer cell lines, i.e., Hep3B, A549, and MCF-7. Compounds **2a**–**c**, **2f**, **2g**, **2j**, **2k**, **3g**, and **3i** were moderate inhibitors against MCF-7.

## 1. Introduction

Thiosemicarbazone derivatives, which play an important role in organic and medicinal chemistry, have attracted a large number of researchers over the years because of their promising biological activities, such as antioxidant [1,2], antiparasitic [3,4,5,6], anticonvulsant [7], antiviral [8,9,10]. Especially, in recent years, large of publications focused on synthesis and antimicrobial [11,12,13,14,15], anticancer [12,14,16,17,18,19,20] activity of thiosemicarbazones to find out the potential candidates to develop new drugs.

Since the first adamantane derivative, amantadine, was found to have antiviral [21,22,23] activity, the synthesis and biological activities of adamantane derivatives, which became an interesting topic, were pursued by a lot of researchers. Therefore, many adamantane derivatives were discovered with various biological activities, i.e., antiviral [24,25,26,27,28,29], antimicrobial [27,29,30,31,32,33,34,35,36,37,38,39], anticancer [32,36,40,41,42,43,44] activities.

To continue to find out the modern molecules, which possess potential bioactivities and contain adamantane skeleton following our previous research [45], herein, we combined two moieties, thiosemicarbazone constituent and adamantane skeleton, which potentially help to confer new molecules with promising biological activities. In this publication, we report the synthesis and biological activities of thiosemicarbazones containing adamantane skeleton.

## 2. Results and Discussion

### 2.1. Chemistry

In this study, 4-(1-adamantyl)-3-thiosemicarbazide (**1**) was used as the key initial material. The condensation of **1** and substituted benzaldehydes under the catalysis of acetic acid and reflux condition in MeOH to yield thiosemicarbazones **2a**–**k** (Scheme 1). Similarly, compound **1** was condensed with substituted acetophenones to yield thiosemicarbazones **3a**–**j** as showed in Scheme 1 and Table 1. The structure of synthesized **2a**–**k** and **3a**–**j** was confirmed by nuclear magnetic resonance, including ^1^H-NMR, ^13^C-NMR, and electron impact (ESI-MS) mass spectral data.

### 2.2. In Vitro Antimicrobial Activity

The synthesized thiosemicarbazones **2a**–**k** and **3a**–**j** were evaluated for their in vitro growing inhibition against seven strains of the National Institute for Food Control (NIFC, Hanoi, Vietnam), including three Gram-negative bacteria (*Escherichia coli* ATCC25922, *Pseudomonas aeruginosa* ATCC27853, and *Salmonella enterica* ATCC12228); three Gram-positive bacteria (*Enterococcus faecalis* ATCC13124, *Staphylococcus aureus* ATCC25923, and *Bacillus cereus* ATCC 13245); and one yeast-like pathogenic fungus, *Candida albicans* ATCC10231. The primary screening was conducted using the microplate dilution method, which utilized Luria-Bertani (LB) broth medium. Streptomycin (an antibiotic) and cycloheximide (an antifungal agent) were used as the positive samples. The results of preliminary antimicrobial testing of the synthesized thiosemicarbazones **2a**–**k** and **3a**–**j** are presented in Table 2 and Table 3.

The results of antimicrobial testing showed that all synthesized thiosemicarbazones variously and seriously inhibited the tested Gram-positive bacteria and *Candida albicans*. Among them, compounds **2c**, **2d**, **2g**, **2j** and **3a**, **3e**, **3g** possessed good on inhibition against EF with a MIC (Minimal inhibitory concentration) of no more than 25 μM and IC_50_ values of 5.68, 4.89, 6.78, 6.88 and 6.34, 4.67 and 4.78 μM, respectively. In term of the inhibition against SA, compounds **2a**, **2e**, **2h**, **2k**, and **3j** showed acceptable activity, with IC_50_ values of 4.78, 6.78, 6.24, 6.45 and 6.46 μM, respectively. Moreover, compounds **2a**, **2e,** and **2g** showed inhibitory effects on BC with IC_50_ values of 4.12, 6.09 and 6.88 μM, individually. Remarkably, all synthesized thiosemicarbazones **2a**–**k** containing adamantane skeletons have significant inhibitory activity against CA. Nevertheless, only compounds **2e** and **2h** had been found to have mild inhibitory effect on PA, a Gram-negative bacterium.

Glancing at the structure-antimicrobial activity relationship of thiosemicarbazones **2a**–**k**, inhibition against EF increased significantly for compounds possessing a (*meta*) –OH group (compounds **2d** and **2h**) or (*para*) –Cl group (compounds **2g** and **2j**) on the phenyl ring. However, the inhibitory activity against SA, BC, EC seem to decrease in the case of substituents on the phenyl ring. Otherwise, inhibition against CA was improved with the presence of substituents on the phenyl ring.

In the series of thiosemicarbazones **3a**–**k**, inhibitory activity against EF increased with the presence of only (*para*) –NO_2_ (compound **3e**) or (*para*)–Cl (compound **3g**) groups alone on the phenyl ring. In case of SAs, the inhibition was improved by having both (*meta*) –NO_2_ and (*para*) –Cl groups (compound **3j**) on the phenyl ring. The inhibition against CA seems to increase when there is a –NO_2_ at position 4 on the phenyl ring (compound **3e**). Among all synthesized thiosemicarbazones **2a**–**k** and **3a**–**j**, only compounds **2d** and **2h** showed the inhibition against PA in all tested Gram-negative bacteria but with limited activity.

### 2.3. In Vitro Cytotoxicity

The synthesized thiosemicarbazones **2a**–**k** and **3a**–**j** were tested for their cytotoxicity in four human cancer cell lines, including Hep3B, HeLa, A549, and MCF-7, in line with a previous publication [46]. As the resulta in Table 4 show, some of the synthesized thiosemicarbazones exhibited antiproliferative activity against the tested cancer cell lines. Among them, compound **2d** showed the killing availability against the tested cell lines with cell viabilities of 16.82 ± 1.60%, 24.55 ± 1.85, 18.37 ± 1.75, and 20.17 ± 1.52 in Hep3B, Hela, A549, and MCF-7 cell lines at a 100 μM dose, respectively. Moreover, compound **2h** was also seen to have good inhibitory effect on the tested cancer cell lines, with cell viabilities of 21.86 ± 0.20%, 34.76 ± 1.36%, 23.86 ± 0.22%, 28.55 ± 1.12% in Hep3B, Hela, A549, and MCF-7 cell lines at a 100 μM dose, respectively. The ability to kill Hep3B, A549, and MCF-7 cancer cell lines of compounds **2d** and **2h** was stronger than that of camptothecin at the same concentration. Additionally, some synthesized compounds showed acceptable inhibition against the growth of MCF-7 such as compounds **2a**–**c**, **2f**, **2g**, **2j**, **2k**, **3g**, and **3i** with viability values of 49.35 ± 0.79%, 43.31 ± 2.63%, 49.02 ± 1.18%, 39.35 ± 1.67%, 33.40 ± 0.86%, 45.33 ± 2.40%, 42.09 ± 0.40%, 46.67 ± 1.49% at a concentration of 100 μM, respectively. Other compounds were seen to have a mild cytotoxicity effect on the tested cancer cell lines. The structure–cytotoxic activity relationship analysis revealed that compounds **2d** and **2h**, which contain (*ortho*) –OH groups on the phenyl ring, have significantly improved killing availability on the tested cancer cell lines. However, an in depth study of the mechanism of the structure–cytotoxicity relationship should be undertaken.

## 3. Experimental

### 3.1. General Information

In this study, 4-(1-adamantyl)-3-thiosemicarbazide (**1**) was synthesized by the reaction of adamantan-1-yl isothiocyanate (Sigma Aldrich, MO, USA) with 80% hydrazine hydrate solution (Sigma Aldrich, St. Louis, MO, USA) in toluene under microwave irradiated conditions for 30 min at 100 W following the same method as a previous publication [47]. Common chemicals such as methanol (MeOH), acetic a**c**id (AcOH), substituted benzaldehydes and acetophenones were also supplied by Sigma Aldrich (St. Louis, MO, USA), Serva Electrophoresis GmbH (Heidelberg, Germany), Fisher Sciencetific (Loughborough, UK), Acros Organics (Branchburg, NJ, USA); and used directly without further purification.

Melting points (°C) were checked in a micro tube glass with an electrothermal melting point apparatus (MP50 Mettler Toledo, Columbus, OH, USA). NMR spectra data was recorded at 500 MHz in DMSO-*d*_6_ using an AVANCE III spectrometer (Bruker Biospin, Billerica, MA, USA); the chemical shifts (δ, ppm) were expressed and coupling constants (*J*) were given in Hz using the internal standard tetramethylsilane. Mass spectra (MS) was obtained on a 910-TQ-FT-MS system, (Agilent, Santa Clara, CA, USA). Processing the reactions and preliminary purity evaluating of synthesized compounds were verified by thin layer chromatography (TLC, pre-coated aluminum sheet 60 F254 plates, Merck KGaA Co., Darmstadt, Germany) and visualization at UV 254 nm. The bacterial and fungus strains, i.e., *Escherichia coli* ATCC25922, *Pseudomonas aeruginosa* ATCC27853, *Salmonella enterica* ATCC12228, *Enterococcus faecalis* ATCC13124, *Stapphylococus aureus* ATCC25923, *Bacillus cereus* ATCC 13245, and *Candida albicans* ATCC10231 were purchased by National Institute for Food Control (NIFC, Hanoi, Vietnam). This study was conducted on cancer cell lines supplied by Advanced Center for Bioorganic Chemistry (ACBC) of the Institute of Marine Biochemistry (IMBC), Vietnam Academy Science and Technology (VAST), i.e., hepatic cancer cell line Hep3B, human cervical cancer cell line HeLa, human lung cancer cell line A549, and human breast carcinoma MCF-7.

### 3.2. Synthesis of Thiosemicarbazones ***2a**–**k*** and ***3a**–**j***

A mixture of 1 mmol of 4-(1-adamantyl)-3-thiosemicarbazide (**1**) and 3 mmol of the appropriate aldehyde or ketone and 20 mL MeOH was put in a three necked glass round bottomed flask and mildly heated. Afterwards glacial acetic acid was dropped into this solution to adjust the pH to 4–5, and the reaction mixture was refluxed. The reaction was monitored by TLC (chloroform/aceton, 95/5, v/v), and visualized under UV at 254 nm and Dragendorff reagent. The reaction was continued until 4-(1-adamantyl)-3-thiosemicarbazide (**1**) was completely consumed. After the reaction was complete, the reaction mixture was evaporated to yield a solid which was washed with ice-cold MeOH to remove residual aldehydes or ketones, and recrystallized from MeOH. The crystals were dried below 50 °C to obtain thiosemicarbazones **2a**–**k** and **3a**–**j.**

*4-(N-Adamantan-1-yl)-1-(1-benzylidene)thiosemicarbazone* (**2a**): ^1^H-NMR (δ ppm): 11.29 (1H, s, -CS-NH-N); 8.08 (1H, s, N=CH); 7.66 (2H, dd, *J1* = 2.0 Hz, *J2* = 7.5 Hz, Ar-H); 7.49 (1H, s, C-NH-CS); 7.44-7.41 (3H, m, Ar-H); 2.28 (6H, m, adamantane-H); 2.08 (3H, m, adamantane-H); 1.66 (6H, m, adamantane-H); ^13^C-NMR (δ ppm): 174.6 (1C, CS); 141.4 (1C, CH=N); 133.8 (1C, Ar-C); 129.9 (1C, Ar-C); 128.8 (2C, Ar-C); 126.9 (2C, Ar-C); 53.0 (1C, C-N); 40.9 (3C, Adamantane-C); 35.9 (3C, Adamantane-C); 29.0 (3C, Adamantane-C). ESI-HR-MS (*m*/*z*): [M + H]^+^ = 314.1676.

*4-(N-Adamantan-1-yl)-1-[1-(3-nitrobenzylidene)]thiosemicarbazone* (**2b**): ^1^H-NMR (δ ppm): 11.50 (1H, s, -CS-NH-N); 8.43 (1H, t, *J1* = 1.5 Hz, *J2* = 8.5 Hz, N=CH); 8.23-8.21 (1H, dd, *J1* = 2.0 Hz, *J2* = 7.5 Hz, Ar-H); 8.18-8.15 (2H, m, Ar-H); 7.73-7.70 (1H, t, *J1* = 8.0 Hz, *J2* = 16.0 Hz, Ar-H); 7.56 (1H, s, C-NH-CS); 2.29 (6H, m, adamantane-H); 2.09 (3H, m, adamantane-H); 1.67 (6H, m, adamantane-H). ^13^C-NMR (δ ppm): 174.8 (1C, CS); 148.2 (1C, Ar-C); 139.0 (1C, CH=N); 135.8 (1C, Ar-C); 132.7 (1C, Ar-C); 130.3 (1C, Ar-C); 124.0 (1C, Ar-C); 121.3 (1C, Ar-C); 53.2 (1C, C-N); 40.7 (3C, adamantane-C); 35.9 (3C, adamantane-C); 29.0 (3C, adamantane-C). ESI-MS (*m*/*z*): [M + H]^+^ = 358.9; [M − H]^−^ = 356.9.

*4-(N-Adamantan-1-yl)-1-[1-(4-methoxybenzylidene)]thiosemicarbazone* (**2c**): ^1^H-NMR (δ ppm): 11.16 (1H, s, -CS-NH-N); 8.02 (1H, s, N=CH); 7.62-7.60 (2H, d, *J* = 9.0 Hz, Ar-H); 7.44 (1H, s, C-NH-CS); 6.99-6.97 (2H, d, *J* = 8.5 Hz, Ar-H); 3.79 (3H, s, OCH_3_); 2.27 (6H, m, adamantane-H); 2.08 (3H, m, adamantane-H); 1.67 (6H, m, adamantane-H). ^13^C-NMR (δ ppm): 174.4 (1C, CS); 160.7 (1C, C4); 141.4 (1C, CH=N); 128.5 (2C, Ar-C); 126.4 (1C, Ar-C); 114.3 (2C, Ar-C); 55.2 (1C, OCH_3_); 52.9 (1C, C-N); 41.0 (3C, adamantane-C); 35.9 (3C, adamantane-C); 29.0 (3C, adamantane-C). *ESI-MS* [*m*/*z*]: [M + H]^+^ = 343.9; [M − H]^−^ = 341.9.

*4-(N-Adamantan-1-yl)-1-[1-(2-hydroxybenzylidene)]thiosemicarbazone* (**2d**): ^1^H-NMR (δ ppm): 11.23 (1H, s, -CS-NH-N); 9.97 (1H, s, OH); 8.37 (1H, s, N=CH); 7.67 (1H, d, *J* = 7.5 Hz, Ar-H); 7.46 (1H, s, C-NH-CS); 7.23-7.20 (1H, t, *J1* = 7.5Hz; *J2* = 15.0 Hz, Ar-H); 6.88-6.87 (1H, d*, J* = 7.5 Hz, Ar-H); 6.85-6.82 (1H, t, *J1* = 7.5 Hz; *J2* = 15.0 Hz, Ar-H); 2.27 (6H, m, adamantane-H); 2.07 (3H, m, adamantane-H); 1.66 (6H, m, adamantane-H). ^13^C-NMR (δ ppm): 174.5 (1C, CS); 156.5 (1C, Ar-C); 138.5 (1C, CH=N); 131.0 (1C, Ar-C); 125.9 (1C, Ar-C); 120.3 (1C, Ar-C); 119.3 (1C, Ar-C); 116.1 (1C, Ar-C); 52.9 (1C, C-N); 41.0 (3C, adamantane-C); 35.9 (3C, adamantane-C); 29.0 (3C, adamantane-C). ESI-MS [*m*/*z*]: [M + H]^+^ = 330.0; [M − H]^−^ = 327.9.

*4-(N-Adamantan-1-yl)-1-[1-(4-nitrobenzylidene)]thiosemicarbazone* (**2e**): ^1^H-NMR (δ ppm): 11.56 (1H, s, -CS-NH-N); 8.25-8.23 (2H, d, *J* = 8.5 Hz, Ar-H); 8.15 (1H, s, N=CH); 7.95-7.93 (2H, d, *J* = 9.0 Hz, Ar-H); 7.57 (1H, s, C-NH); 2.29 (6H, m, adamantane-H); 2.09 (3H, m, adamantane-H); 1.66 (6H, m, adamantane-H). ^13^C-NMR (δ ppm): 174.8 (1C, CS); 147.6 (1C, Ar-C); 140.3 (1C, CH=N); 138.8 (1C, Ar-C); 127.8 (2C, Ar-C); 123.9 (2C, Ar-C); 53.3 (C-NH); 40.7 (3C, adamantane-C); 35.9 (3C, adamantane-C); 29.0 (3C, adamantane-C). ESI-MS [m/z]: [M + H]^+^ = 358.9; [M − H]^−^ = 356.9.

*4-(N-Adamantan-1-yl)-1-[1-(4-ethoxybenzylidene)]thiosemicarbazone* (**2f**): ^1^H-NMR (δ ppm): 11.16 (1H, s, -CS-NH-N); 8.01 (1H, s, N=CH); 7.60-7.58 (2H, d, *J* = 8.6 Hz, Ar-H); 7.43 (1H, s, C-NH); 6.97-6.95 (2H, d, *J* = 8.5; Ar-H); 4.07-4.05 (2H, d, *J* = 7.0 Hz, OCH_2_); 2.27 (6H, m, adamantane-H); 2.08 (3H, m, adamantane-H); 1.66 (6H, m, adamantane-H); 1.35-1.32 (3H, t, *J1* = 7.0 Hz, *J2* = 14.0 Hz, CH_2_-CH_3_). ^13^C-NMR (δ ppm): 174.4 (1C, CS); 160.0 (1C, Ar-C); 141.5 (1C, CH=N); 128.5 (2C, Ar-C); 126.2 (1C, Ar-C); 114.7 (2C, Ar-C); 63.2 (1C, OCH_2_); 52.8 (C-NH); 41.0 (3C, adamantane-C); 35.9 (3C, adamantane-C); 29.0 (3C, adamantane-C); 14.5 (1C, CH_2_CH_3_). *ESI-MS* [*m*/*z*]: [M + H]^+^ = 358.0; [M − H]^−^ = 355.9.

*4-(N-Adamantan-1-yl)-1-[1-(4-chlorobenzylidene)]thiosemicarbazone* (**2g**): ^1^H-NMR (δ ppm): 11.34 (1H, s, -CS-NH-N); 8.05 (1H, s, N=CH); 7.70 (2H, d, *J* = 8.5 Hz, Ar-H); 7.48-7.47 (3H, m, Ar-H & NH-C); 2.28 (6H, m, adamantane-H); 2.08 (3H, m, adamantane-H); 1.66 (6H, m, adamantane-H). ^13^C-NMR (δ ppm): 174.7 (1C, CS); 140.1 (1C, CH=N); 134.3 (1C, Ar-C); 132.8 (1C, Ar-C); 128.8 (2C, Ar-C); 128.6 (2C, Ar-C); 53.1 (C-NH); 40.9 (3C, adamantane-C); 35.9 (3C, adamantane-C); 29.0 (3C, adamantane-C). *ESI-MS* [*m*/*z*]: [M + H]^+^ = 347.9; [M − H]^−^ = 345.9.

*4-(N-Adamantan-1-yl)-1-[1-(2-hydroxy-5-methylbenzylidene)]thiosemicarbazone* (**2h**): ^1^H-NMR (δ ppm): 11.20 (1H, s, -CS-NH-N); 9.73 (1H, s, OH); 8.34 (1H, s, N=CH); 7.45 (1H, s, C-NH); 7.44 (1H, s, Ar-H); 7.04-7.02 (1H, dd, *J1* = 2.0 Hz, *J2* = 8.5 Hz, Ar-H); 6.78-6.76 (1H, d, *J* = 7.5 Hz, Ar-H); 2.27 (6H, m, adamantane-H); 2.21 (3H, s, CH_3_); 2.08 (3H, m, adamantane-H); 1.66 (6H, m, adamantane-H). ^13^C-NMR (δ ppm): 174.5 (1C, CS); 154.4 (1C, Ar-C); 138.9 (1C, CH=N); 131.8 (1C, Ar-C); 127.8 (1C, Ar-C); 125.9 (1C, Ar-C); 119.8 (1C, Ar-C); 116.0 (1C, Ar-C); 52.8 (C-NH); 40.9 (3C, adamantane-C); 35.9 (3C, adamantane-C); 29.0 (3C, adamantane-C); 20.1 (1C, CH_3_). ESI-MS [*m*/*z*]: [M + H]^+^ = 344.0; [M − H]^−^ = 341.9.

*4-(N-Adamantan-1-yl)-1-[1-(3-nitro-4-ethoxybenzylidene)]thiosemicarbazone* (**2i**): ^1^H-NMR (δ ppm): 11.34 (1H, s, -CS-NH-N); 8.16-8.15 (1H, d, *J* = 2.5 Hz, Ar-H); 8.05 (1H, s, N=CH); 7.93-7.91 (1H, dd, *J1* = 2.0 Hz, *J2* = 8.5 Hz, Ar-H); 7.48 (1H, s, C-NH); 7.39-7.37 (1H, d, *J* = 8.5 Hz, Ar-H); 4.28-4.24 (2H, dd, *J1* = 7.0 Hz; *J2* = 14.0 Hz, OCH_2_); 2.28 (6H, m, adamantane-H); 2.08 (3H, m, adamantane-H); 1.66 (6H, m, adamantane-H); 1.36-1.33 (3H, t, *J1* = 7.0; *J2* = 14.0 Hz, CH_3_). ^13^C-NMR (δ ppm): 174.6 (1C, CS); 151.7 (1C, Ar-C); 140.0 (1C, CH=N); 139.2 (1C, Ar-C); 132.2 (1C, Ar-C); 126.5 (1C, Ar-C); 122.8 (1C, Ar-C); 115.3 (1C, Ar-C); 65.3 (1C, OCH_2_); 53.1 (C-NH); 40.8 (3C, adamantane-C); 35.9 (3C, adamantane-C); 29.0 (3C, adamantane-C); 14.2 (1C, CH_3_). ESI-MS [*m*/*z*]: [M + H]^+^ = 403.0; [M − H]^−^ = 400.9.

*4-(N-Adamantan-1-yl)-1-[1-(3-nitro-4-chlorobenzylidene)]thiosemicarbazone* (**2j**): ^1^H-NMR (δ ppm): 11.53 (1H, s, -CS-NH-N); 8.36 (1H, d, *J* = 1.0 Hz, N=CH); 8.09 (1H, s, Ar-H); 8.0 (1H, d, *J* =8.0 Hz, Ar-H); 7.79 (1H, d, *J* = 8.5 Hz, Ar-H); 7.54 (1H, s, NH-C); 2.28 (6H, m, adamantane-H); 2.08 (3H, m, adamantane-H); 1.66 (6H, m, adamantane-H). ^13^C-NMR (δ ppm): 174.8 (1C, CS); 148.1 (1C, Ar-C); 137.9 (1C, CH=N); 134.7 (1C, Ar-C); 131.8 (1C, Ar-C); 131.3 (1C, Ar-C); 125.0 (1C, Ar-C); 123.2 (1C, Ar-C); 53.3 (C-NH); 40.7 (3C, adamantane-C); 35.9 (3C, adamantane-C); 29.0 (3C, adamantane-C). ESI-MS [*m*/*z*]: [M + H]^+^ = 392.9; [M − H]^−^ = 390.9.

*4-(N-Adamantan-1-yl)-1-[1-(2,5-dimethylbenzylidene)]thiosemicarbazone* (**2k**): ^1^H-NMR (δ ppm): 11.21 (1H, s, -CS-NH-N); 8.33 (1H, s, N=CH); 7.49 (1H, s, Ar-H); 7.42 (1H, NH-C); 7.12-7.11 (2H, m, Ar-H); 2.35 (3H, s, CH_3_); 2.29 (3H, s, CH_3_); 2.27 (6H, m, adamantane-H); 2.07 (3H, m, adamantane-H); 1.66 (6H, m, adamantane-H). ^13^C-NMR (δ ppm): 174.5 (1C, CS); 141.2 (1C, CH=N); 135.1 (1C, Ar-C); 133.7 (1C, Ar-C); 131.5 (1C, Ar-C); 130.9 (1C, Ar-C); 130.3 (1C, Ar-C); 127.0 (1C, Ar-C); 52.9 (C-NH); 40.9 (3C, adamantane-C); 35.8 (3C, adamantane-C); 29.0 (3C, adamantane-C); 20.4 (1C, CH_3_); 19.2 (1C, CH_3_). ESI-MS [*m*/*z*]: [M + H]^+^ = 342.0; [M − H]^−^ = 340.0.

*4-(N-Adamantan-1-yl)-1-(1-phenylethylidene)thiosemicarbazone* (**3a**): ^1^H-NMR (δ ppm): 9.77 (1H, s, -CS-NH-N); 7.69 (2H, d, *J* = 8.5 Hz, Ar-H); 7.63 (1H, s, Ar-H); 7.59 (2H, d, *J* = 8.5 Hz, Ar-H); 2.30 (3H, s, CH_3_); 2.28 (6H, s, adamantane-H); 2.09 (3H,s, adamantane-H); 1.68 (6H, s, adamantane-H). ^13^C-NMR (δ ppm): 175.5 (1C, CS); 147.1 (1C, N=C); 137.7 (1C, Ar-C); 129.2 (1C, Ar-C); 128.4 (2C, Ar-C); 126.0 (2C, Ar-C); 52.9 (C-NH); 40.8 (3C, adamantane-C); 35.8 (3C, adamantane-C); 28.9 (3C, adamantane-C); 14.2 (1C, CH_3_). ESI-MS [*m*/*z*]: [M + H]^+^ = 325.9; [M − H]^−^ = 328.0.

*4-(N-Adamantan-1-yl)-1-[1-(3-nitrophenyl)ethylidene]thiosemicarbazone* (**3b**): ^1^H-NMR (δ ppm): 10.22 (1H, s, -CS-NH-N); 8.49 (1H, s, Ar-H); 8.23 (1H, dd, *J1* = 1.5 Hz, *J2* = 8.0 Hz, Ar-H); 8.19 (1H, d, *J* = 8.0 Hz, Ar-H); 7.73 (1H, s, NH-C); 7.70 (1H, d, *J* = 8.0 Hz, Ar-H); 2.37 (3H, s, CH_3_); 2.28 (6H, brs, adamantane-H); 2.08 (3H, brs, adamantane-H); 1.66 (6H, brs, adamantane-H). ^13^C-NMR (δ ppm): 175.7 (1C, CS); 148.1 (1C, N=C); 144.8 (1C, Ar-C); 139.5 (1C, Ar-C); 132.4 (1C, Ar-C); 130.2 (1C, Ar-C); 123.6 (1C; Ar-C); 120.5 (1C; Ar-C); 53.2 (C-NH); 40.8 (3C, adamantane-C); 35.9 (3C, adamantane-C); 29.0 (3C, adamantane-C); 14.3 (1C, CH_3_). ESI-MS [*m*/*z*]: [M + H]^+^ = 372.9; [M − H]^−^ = 370.9.

*4-(N-Adamantan-1-yl)-1-[1-(4-bromophenyl)ethylidene]thiosemicarbazone* (**3c**): ^1^H-NMR (δ ppm): 9.77 (1H, s, -CS-NH-N); 7.70-7.68 (2H, d, *J* = 8.5 Hz, Ar-H); 7.63 (1H, s, NH-C); 7.60-7.59 (2H, d, *J* = 8.5 Hz, Ar-H); 2.29 (3H, s, CH_3_); 2.28 (3H, s, CH_3_); 2.09 (3H, m, adamantane-H); 1.68 (6H, m, adamantane-H). ^13^C-NMR (δ ppm): 175.5 (1C, CS); 146.2 (1C, C=N); 136.9 (1C, Ar-C); 131.4 (2C, Ar-C); 128.2 (2C, Ar-C); 122.7 (1C, Ar-C); 53.1 (C-NH); 40.8 (3C, adamantane-C); 35.9 (3C, adamantane-C); 29.0 (3C, adamantane-C); 14.1 (1C, CH_3_). ESI-MS [*m*/*z*]: [M + H]^+^ = 407.9; [M − H]^−^ = 405.8.

*4-(N-Adamantan-1-yl)-1-[1-(4-hydroxyphenyl)ethylidene]thiosemicarbazone* (**3d**): ^1^H-NMR (δ ppm): 9.83 (1H, s, -CS-NH-N); 7.64 (1H, s, NH-C); 7.59 (2H, d, *J* = 8.5 Hz, Ar-H); 6.79 (2H, d, *J* = 9.0 Hz, Ar-H); 2.26 (6H, brs, adamantane-H); 2.24 (3H, s, CH_3_); 2.07 (3H, brs, adamantane-H); 1.65 (6H, brs, adamantane-H). ^13^C-NMR (δ ppm): 175.3 (1C, CS); 158.8 (1C, Ar-C), 147.7 (1C, N=C); 128.4 (1C, Ar-C); 127.7 (2C, Ar-C); 115.3 (2C, Ar-C); 52.9 (C-NH); 41.0 (3C, adamantane-C); 35.9 (3C, adamantane-C); 29.0 (3C, adamantane-C); 14.1 (1C, CH_3_). ESI-MS [*m*/*z*]: [M + H]^+^ = 344.0; [M − H]^−^ = 341.9.

*4-(N-adamantan-1-yl)-1-[1-(4-nitro-phenyl)ethylidene]thiosemicarbazone* (**3e**): ^1^H-NMR (500 MHz, DMSO-*d_6_*, δ ppm): 10.25 (1H, s, -CS-NH-N); 8.24 (2H, d, *J* = 9.0 Hz, Ar-H); 8.01 (2H, d, J = 9.0 Hz, Ar-H); 7.72 (1H, s, NH-C); 2.36 (3H, s, CH_3_); 2.28 (6H, brs, adamantane-H); 2.08 (3H, brs, adamantane-H); 1.66 (6H, brs, adamantane-H). ^13^C-NMR (125 MHz, DMSO-*d_6_*, δ ppm): 175.6 (1C, CS); 147.5 (1C, Ar-C), 144.8 (1C, N=C); 144.0 (1C, Ar-C); 127.3 (2C, Ar-C); 123.6 (2C, Ar-C); 53.3 (C-NH); 40.7 (3C, adamantane-C); 35.9 (3C, adamantane-C); 29.0 (3C, adamantane-C); 14.3 (1C, CH_3_). ESI-MS [*m*/*z*]: [M + H]^+^ = 372.9; [M − H]^−^ = 370.9.

*4-(N-Adamantan-1-yl)-1-[1-(3-nitro-4-bromo-phenyl)ethylidene]thiosemicarbazone* (**3f**): ^1^H-NMR (500 MHz, DMSO-*d_6_*, δ ppm): 10.22 (1H, s, -CS-NH-N); 8.33 (1H, d, *J* = 1.5 Hz, Ar-H); 7.96-7.92 (2H, m, Ar-H); 7.69 (1H, s, NH-C); 2.33 (3H, s, CH_3_); 2.28 (6H, s, adamantane-H); 2.08 (3H, brs, adamantane-H); 1.66 (6H, brs, adamantane-H). ^13^C-NMR (125 MHz, DMSO-*d_6_*, δ ppm): 175.5 (1C, CS); 149.9 (1C, Ar-C); 143.9 (1C, N=C); 138.7 (1C; Ar-C); 134.6 (1C; Ar-C); 130.7 (1C; Ar-C); 122.7 (1C; Ar-C); 113.1 (1C, Ar-C); 53.2 (C-NH); 40.6 (3C, adamantane-C); 35.8 (3C, adamantane-C); 28.9 (3C, adamantane-C); 14.0 (1C, CH_3_). ESI-MS [*m*/*z*]: [M + H]^+^ = 452.9; [M − H]^−^ = 450.8.

*4-(N-Adamantan-1-yl)-1-[1-(4-chloro-phenyl)ethylidene]thiosemicarbazone* (**3g**): ^1^H-NMR (500 MHz, DMSO-*d_6_*, δ ppm): 10.07 (1H, s, -CS-NH-N); 7.77 (2H, d, *J* = 8.5 Hz, Ar-H); 7.67 (1H, s, NH-C); 7.47 (2H, d, *J* = 8.5 Hz, Ar-H); 2.30 (3H, s, CH_3_); 2.27 (6H, s, adamantane-H); 2.08 (3H, s, adamantane-H); 1.66 (6H, s, adamantane-H). ^13^C-NMR (125 MHz, DMSO-*d_6_*, δ ppm): 175.5 (1C, CS); 146.0 (1C, N=C); 136.5 (1C, Ar-C); 133.9 (1C, Ar-C); 128.4 (2C, Ar-C); 127.9 (2C, Ar-C); 53.0 (C-NH); 40.8 (3C, adamantane-C); 35.8 (3C, adamantane-C); 28.9 (3C, adamantane-C); 14.1 (1C, CH_3_). ESI-MS [*m*/*z*]: [M + H]^+^ = 361.9; [M − H]^−^ = 359.9.

*4-(N-Adamantan-1-yl)-1-[1-(4-methylphenyl)ethylidene]thiosemicarbazone* (**3h**): ^1^H-NMR (δ ppm): 9.69 (1H, s, -CS-NH-N); 7.64-7.64 (3H, m, Ar-H and NH-C); 7.22 (2H, d, *J* = 7.0 Hz, Ar-H); 2.33 (3H, s, CH_3_); 2.29 (9H, s, CH_3_ and adamantane-H); 2.09 (3H, s, adamantane-H); 1.68 (6H, s, adamantane-H). ^13^C-NMR (δ ppm): 175.6 (1C, CS); 147.1 (1C, N=C); 138.7 (1C, Ar-C); 134.9 (1C, Ar-C); 128.9 (2C, Ar-C); 125.8 (2C, Ar-C); 52.9 (C-NH); 40.9 (3C, adamantane-C); 35.8 (3C, adamantane-C); 28.9 (3C, adamantane-C); 20.5 (1C, CH_3_); 13.8 (1C, CH_3_). ESI-MS [*m*/*z*]: [M + H]^+^ = 342.0.

*4-(N-Adamantan-1-yl)-1-[1-(3-nitro-4-methoxyphenyl)ethylidene]thiosemicarbazone* (**3i**). ^1^H-NMR (δ ppm): 9.83 (1H, s, -CS-NH-N); 8.18 (1H, d, *J* = 2.0 Hz, Ar-H); 8.02 (1H, dd, *J1* = 2.0 Hz, *J2* = 8.5 Hz, Ar-H); 7.64 (1H, s, NH-C); 7.40 (1H, d, *J* = 9.0 Hz, Ar-H); 2.32 (3H, s, CH_3_); 2.29 (6H, s, adamantane-H); 2.10 (3H, s, adamantane-H); 1.68 (6H, s, adamantane-H). ^13^C-NMR (δ ppm): 175.4 (1C, CS); 152.3 (1C, Ar-C); 144.9 (1C; N=C); 139.3 (1C; Ar-C); 131.6 (1C, Ar-C); 130.1 (1C, Ar-C); 122.4 (1C, Ar-C); 114.3 (1C, Ar-C); 53.0 (C-NH); 40.7 (3C, adamantane-C); 35.9 (3C, adamantane-C); 28.9 (3C, adamantane-C); 14.0 (1C, CH_3_). ESI-MS [m/z]: [M + H]^+^ = 403.0; [M − H]^−^ = 400.9.

*4-(N-Adamantan-1-yl)-1-[1-(3-nitro-4-chlorophenyl)ethylidene]thiosemicarbazone* (**3j**). ^1^H-NMR (δ ppm): 10.22 (1H, s, -CS-NH-N); 8.38 (1H, d, *J* = 1.5 Hz, Ar-H); 8.07-8.05 (1H, dd, *J1* = 1.5 Hz, *J2* = 8.5 Hz, Ar-H); 7.8 (1H, d, *J* = 8.5 Hz, Ar-H); 7.69 (1H, s, NH-C); 2.33 (3H, s, CH_3_); 2.28 (3H, s, adamantane-H); 2.08 (3H, s, adamantane-H); 1.66 (6H, s, adamantane-H). ^13^C-NMR (δ ppm): 175.5 (1C, CS); 147.8 (1C, Ar-C); 143.8 (1C, C=N); 138.2 (1C, Ar-C); 131.6 (1C, Ar-C); 130.8 (1C, Ar-C); 124.9 (1C, Ar-C); 122.8 (1C, Ar-C); 53.2 (C-NH); 40.6 (3C, adamantane-C); 35.9 (3C, adamantane-C); 28.9 (3C, adamantane-C); 14.0 (1C, CH_3_). ESI-MS [*m*/*z*]: [M + H]^+^ = 406.9; [M − H]^−^ = 404.9.

Spectra of the synthesized compounds can be found in the Appendix A.

### 3.3. Dertemination of Antimicrobial Activity by the Dilution Method 

The synthesized compounds **2a**–**k**, **3a**–**j** were dissolved in DMSO at the concentration of 100 mM, separately, to prepare the stock solutions. A total of 0.4 mL of the stock solution was withdrawn and 9.6 mL LB media was added and mixed homogenously to obtain a working solution. Then, twofold serial dilution solutions in LB media was prepared at the concentration from 100 μM to 0.78 μM in 96 well plates, in triplicate. Before being incubating at 37 °C for 24 h, 50 μL of test microorganism suspension at 2 × 10^5^ CFU/mL was inoculated to each well. Streptomycin and cycloheximide were used as the positive control samples. Minimum inhibitory concentration (MIC) was defined as the lowest concentration that completely inhibited the development of the microorganism, which was detected by the naked eye. Half of the maximum inhibitory concentration (IC_50_) is defined as the concentration of tested compounds that inhibit 50% visual growth of the test microorganism, which was determined by using turbidity measurement on a iMark™ Microplate Absorbance Reader (Bio-Rad Laboratories, Hercules, CA, USA) [48].

### 3.4. Determination of Cytotoxicity Activity

The MTT (3-(4,5-dimethythiazol-2-yl)-2,5-diphenyltetrazolium bromide) method was used to determine the cytotoxicity of the synthesized compounds **2a**–**k**, **3a**–**j** on human cancer cell lines, i.e., Hep3B (hepatic cancer cell line), HeLa (human cervical cancer cell line), A549 (human lung cancer cell line), and MCF-7 (human breast carcinoma), as described in a previous publication [46]. Tested cells were seeded at 5 × 10^6^ cell/well in 96-well plates with RPMI 1640 or DMEM media containing 10% fetal bovine serum, penicillin (100 IU/mL), and streptomycin (100 μg/mL) at 37 °C in a humid air incubator supplied with 5% CO_2_. The cells were stabilized for 24 h and then were removed to old media and treated with the sample. Each 200 μL of the test samples was added to the well and incubated during 72 h in culture conditions. Following removing the medium, 50 μL MTT solution (1 mg/mL in phosphate buffer saline) was then poured into to each well and the cells continue to be incubated at 37 °C for 4 h. After eliminating the MTT solution, 100 μL of isopropanol was added to each well. The absorbance was measured on a iMark™ Microplate Absorbance Reader (Bio-Rad Laboratories, Hercules, CA, USA) at 570 nm. Suitable blank and positive controls (camptothecin) were included. Cytotoxicity of the synthesized compounds was defined as the percent of cell survival, as follows: [OD (72 h)–OD (0 h)]/[OD(DMSO)–OD (0 h)]; where OD (72 h), OD (0 h), OD(DMSO) are the absorbance of the test sample at 72 h, test sample at 0 h, and the DMSO sample.

## 4. Conclusions

The synthesis and characterization of thiosemicarbazones containing adamantane skeletons **2a**–**k** and **3a**–**j**, were achieved. The syntheses were performed by condensing 4-(1-adamantyl)-3-thiosemicarbazide (**1**) with substituted benzaldehydes to get compounds **2a**–**k**, and with substituted acetophenones to get compounds **3a**–**j**. The antimicrobial and cytotoxicity of the synthesized compounds were determined. The screening results showed that all synthesized thiosemicarbazones have good inhibitory activity against CA. Among them, compounds **2c**, **2d**, **2g**, **2j** and **3a**, **3e**, **3g** displayed the inhibitory activity against EF. Compounds **2a**, **2e**, **2h**, **2k** and **3j** were moderate inhibitors against SA. Compounds **2a**, **2e** and **2g** were found to have so good inhibitory effect on BC. Screening of structure-antimicrobial activity relationship of thiosemicarbazones which were synthesized by condensing 4-(1-adamantyl)-3-thiosemicarbazide (**1**) with substituted benzaldehydes, replacing the –H atom by substituents on the phenyl ring improved the inhibition against EF and CA but decreased that of SA and BC. Nevertheless, in case the compounds were formed by condensing 4-(1-adamantyl)-3-thiosemicarbazide (**1**) with substituted acetophenones, the inhibition against EF, SA, BC and CA seemed to decrease if replacing –H atom by substituents on the phenyl ring (except compounds **3e**, **3g** and **3f**). Among all synthesized thiosemicarbazones, compounds **2d** and **2h**, which contained (*ortho*) hydroxyl groups on the phenyl ring, showed good inhibitory activity against the tested cancer cell lines, i.e., Hep3B, A549, and MCF-7. Moreover, compounds **2a**–**c**, **2f**, **2g**, **2j**, **2k**, **3g**, and **3i** were moderate inhibitors against MCF-7.

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
