# Peer review of "Synthesis and Bioactivity of Thiosemicarbazones Containing Adamantane Skeletons"

_molecules, 2020, doi:10.3390/molecules25020324_

Round 1

Reviewer 1 Report

In this manuscript the authors report on preparation and bioactivity of adamantyl thiosemicarbazone compounds. The authors screened the derivatives against six bacteria strains, one fungus and four human cancer cell lines. However, there are some issues with the manuscript in its present form that need to be addressed before publication.  

1. Significant improvements in the quality of the English language and style should be undertaken so that the paper achieves the expected professional level of a scientific report. A critical reading of the text by a native speaker is therefore highly recommended.

2. Please redraw the adamantane-containing structures on Figure 1, the bond of substituent with the bridgehead adamantane position is at the wrong angle and therefore chemically incorrect.

3. There is no mention in the manuscript about the origin of the starting compound 1. Either list it as a commercially obtained chemical if it was bought or cite the literature regarding its preparation if it was synthesized.

4. A number of adamantyl thiosemicarbazones reported in this manuscript were previously reported in the literature or are even commercially available (2a, 2d, 2e, 2g, etc.). Please cite relevant literature and refrain from using the term novel when describing these known derivatives (e.g., Molecules 2019, 24(23), 4308;etc.). Related to this point, for some of the known compounds bioactivity assessment has been reported as well so please comment on the existing literature results in your paper and compare them with your own findings. Overall, relevant literature on this topic has not been sufficiently covered (e.g., Bioorganic Chemistry 92 (2019) 103244) and I suggest that the authors make efforts to resolve this serious issue.

5. At several places in the text the term “modern synthesized thiosemicarbazones” appears, please rephrase this since it is not clear at all what is meant by it.

6. The authors mention that “Nevertheless, the mechanism of the structure- antimicrobial relationship should be further studied.” and “However, the deep mechanism of structure – cytotoxicity relationship should be undertaken.” relating to antimicrobial and antiproliferative activity results, respectively. How do they propose to do that and what are their future plans in this direction?

In the end, it is my opinion that this manuscript would be of interest to the readership of the journal Molecules. I would therefore recommend it for publication but only after these revisions have been undertaken in full.

Author Response

Respond to Reviewer 1

First, thank you for all your constructive comments about our manuscript. Your valuable suggestions are the best point for helping us to complete our research.
From your comment, we respond and explain as the following:

Comment 1: Significant improvements in the quality of the English language and style should be undertaken so that the paper achieves the expected professional level of a scientific report. A critical reading of the text by a native speaker is therefore highly recommended.

Respond 1: Thank you for your suggestion that helps us complete our manuscript right formal and expert English language to increase the scientific value of our study. From your suggestion, we have already amended our manuscript and asked an English expert to check it.

Comment 2: Please redraw the adamantane-containing structures on Figure 1, the bond of substituent with the bridgehead adamantane position is at the wrong angle and therefore chemically incorrect.

Respond 2: Thank you for your exact comment. From your point, we have already redraw adamantane-containing structures on Figure 1 in exact form.

Comment 3: There is no mention in the manuscript about the origin of the starting compound 1. Either list it as a commercially obtained chemical if it was bought or cite the literature regarding its preparation if it was synthesized.

Respond 3: Thank you for your constructive comment. From your suggestion, we have already written down the origin of compound 1 and other commercially obtained chemicals at titled 3. Experiments (were detailed in 3.1. Summary).

Comment 4: A number of adamantyl thiosemicarbazones reported in this manuscript were previously reported in the literature or are even commercially available (2a, 2d, 2e, 2g, etc.). Please cite relevant literature and refrain from using the term novel when describing these known derivatives (e.g., Molecules 2019, 24(23), 4308;etc.). Related to this point, for some of the known compounds bioactivity assessment has been reported as well so please comment on the existing literature results in your paper and compare them with your own findings. Overall, relevant literature on this topic has not been sufficiently covered (e.g., Bioorganic Chemistry 92 (2019) 103244) and I suggest that the authors make efforts to resolve this serious issue.

Respond 4: Thank you for your great suitable suggestion. From your point, we had already checked some citations that you recommended. It is exactly that some compounds were reported in the recently publication (Molecules 2019, 24(23), 4308), however, we checked the published time of this publication, it was 26 November 2019, very near now, after the time I prepared my manuscript to ask my research leader and my university asking the permission for sending the manuscript to Molecules Journal (on 25 November 2019). It could only be explained that our experiment was conducted at the same time with the report at Molecules 2019, 24(23), 4308. Moreover, all the original data of our manuscript were showed very detail in Supplementary File demonstrated that our study was carried out in a serious and strict manner. Absorbing the suggestion of Reviewer, we amended "novel synthesized thiosemicarbazone" or "newly synthesized thiosemicarbazone" to "synthesized thiosemicarbazone" in all our manuscript.
Besides, Reviewers pointed out that some of the compounds' bioactivity assessment has been reported and suggested us to compare our result to a previous publication (Bioorganic Chemistry 92 (2019) 103244). We would like to thank for your broaden discussion in our manuscript. However, we have already checked this publication and found that the objects in our manuscript differed to
the above publication. The target molecules in our research were 4-(N-adamantan-1-yl)-1-arylthiosemicarbazones but above publication's was 4-(N-aryl)-1-(adamantan-1-yl)thiosemicarbazones. Moreover, in Bioorganic Chemistry 92 (2019) 103244, the tested bioactivity was antidiabetic activity but our research only tested for antimicrobial and cytotoxicity of synthesized compounds. For the two above reasons, we did not compare our research with the result in Bioorganic Chemistry 92 (2019) 103244.

Comment 5: At several places in the text the term “modern synthesized thiosemicarbazones” appears, please rephrase this since it is not clear at all what is meant by it.

Respond 5: Thank you for your recommendation. We have already amended all phrase “modern synthesized thiosemicarbazones” to " synthesized thiosemicarbazones”.

Comment 6: The authors mention that “Nevertheless, the mechanism of the structure- antimicrobial relationship should be further studied.” and “However, the deep mechanism of structure – cytotoxicity relationship should be undertaken.” relating to antimicrobial and antiproliferative activity results, respectively. How do they propose to do that and what are their future plans in this direction?

Respond 6: Thank you for your review. We mention it for our research orientation about these derivatives in the future. However, from the Reviewer's suggestion, we have already removed these emphases for our manuscript more clearly.

Finally, we would like to thank for all your constructive comments. Your valuable suggestions are the best point for helping us to complete our research.

Reviewer 2 Report

Comments:

In the Abstract should be amended to “Reaction of 4-(1-adamantyl)-3-thiosemicarbazide (1) with numerous substituted acetophenones and benzaldehydes yielded the corresponding thiosemicarbazones containing adamantane skeleton. The novel synthesized compounds were evaluated for activities against some Gram-positive and Gram-negative bacteria, and the fungus Candida albicans, and cytotoxicity on four cancer cell lines, including Hep3B, HeLa, A549, and MCF-7. All of them showed good inhibitory effect on Candida albicans. Compounds 2c, 2d, 2g, 2j and 3a, 3e, 3g possessed so well in inhibition against Enterococcus faecalis. Compounds 2a, 2e, 2h, 2k and 3j possibly have inhibition against Staphylococcus aureus and Bacillus cereus with medium acceptance level. Compounds 2a, 2e and 2g found so good inhibitory effect on Bacillus cereus. Compounds 2d and 2h, which contain –OH (ortho) group on phenyl ring, were found as good candidates to kill tested cancer cell lines, i.e., Hep3B, A549, and MCF-7. Some of them such as 2a-c, 2f, 2g, 2j, 2k, 3g, and 3i showed the acceptable effect on the growth of MCF-7.”

Page 3, line 7, in the Table 1, “visualization at UV 254 nm.” should be amended “visualization at UV 254 nm. Rf: retention factor” Page 3, line 16, “Cyloheximide” should be amended “Cycloheximide”. Page 5, line 6, 8, 9, 19, “HeP3B” should be amended “Hep3B”. Page 5, line 16, and Page 10, line 36, “–OH (octo) group” should be amended “–OH (ortho) group”. Page 6, “2. Synthesis of Thiosemicarbazones 2a-k and 3a-j” should be amended “3.2. Synthesis of Thiosemicarbazones 2a-k and 3a-j. Page 6 ~ 9 “13C-NMR (500 MHz, DMSO-d6, δ ppm):” should be amended “13C-NMR (125 MHz, DMSO-d6, δ ppm):”. Page 7, line 6, “1H-NMR (500MHz, DMSO-d6, δ ppm) should be amended “1H-NMR (500 MHz, DMSO-d6, δ ppm).

In the “Supporting information”, page S6-S8 “1a” should be amended “2a”.

Author Response

Respond to Reviewer 2
First, thank you for all your constructive commendation. Your valuable
suggestions are the best point helping us to complete our research.
From your comment, we responded and explained as the following:

Comment 1: In the Abstract should be amended to “Reaction of 4-(1-
adamantyl)-3-thiosemicarbazide (1) with numerous substituted acetophenones
and benzaldehydes yielded the corresponding thiosemicarbazones containing
adamantane skeleton. The novel synthesized compounds were evaluated for
activities against some Gram-positive and Gram-negative bacteria, and the
fungus Candida albicans, and cytotoxicity on four cancer cell lines, including
Hep3B, HeLa, A549, and MCF-7. All of them showed good inhibitory effect
on Candida albicans. Compounds 2c, 2d, 2g, 2jand 3a, 3e, 3g possessed so well
in inhibition against Enterococcus faecalis.
Compounds 2a, 2e, 2h, 2k and 3j possibly have inhibition
against Staphylococcus aureus and Bacillus cereus with medium acceptance
level. Compounds 2a, 2e and 2gfound so good inhibitory effect on Bacillus
cereus. Compounds 2d and 2h, which contain –OH (ortho) group on phenyl ring,
were found as good candidates to kill tested cancer cell lines, i.e., Hep3B, A549,
and MCF-7. Some of them such as 2a-c, 2f, 2g, 2j, 2k, 3g, and 3i showed the
acceptable effect on the growth of MCF-7.”

Respond 1: Thank you for your review. We have already amended it
following the Reviewer's suggestion.

Comment 2: Page 3, line 7, in the Table 1, “visualization at UV 254 nm.”
should be amended “visualization at UV 254 nm. Rf: retention factor” Page 3,
line 16, “Cyloheximide” should be amended “Cycloheximide”. Page 5, line 6, 8,
9, 19, “HeP3B” should be amended “Hep3B”. Page 5, line 16, and Page 10, line
36, “–OH (octo) group” should be amended “–OH (ortho) group”. Page 6, “2.
Synthesis of Thiosemicarbazones 2a-k and 3a-j” should be amended “3.2.
Synthesis of Thiosemicarbazones 2a-k and 3a-j”. Page 6 ~ 9 “13C-NMR (500
MHz, DMSO-d6, δ ppm):” should be amended “13C-NMR (125 MHz, DMSO-d6,
δ ppm):”. Page 7, line 6, “1H-NMR (500MHz, DMSO-d6, δ ppm)”should be
amended “1H-NMR (500 MHz, DMSO-d6, δ ppm)”.

Respond 2: Thank you for your consideration to our manuscript. We have
already rewritten these points following the Reviewer's suggestion.

Comment 3: In the “Supporting information”, page S6-S8 “1a” should be
amended “2a”.

Respond 3: Thank you for suitable suggestion. We have already corrected
this error in our manuscript.

In conclusion, we would like to thank for the constructive commendation
of Reviewer. Your advices are useful points for we complete our manuscript more
clearly and make it be suitable for Molecules Journal.
Thank you very much.

Reviewer 3 Report

The manuscript concerned the synthesis and biological evaluation of adamantane derivatives, functionalized by thiosemicarbazone moiety with differently substituted phenyl aldehydes or methyl phenyl ketones as a screening for their biological activity against different bacteria and citotoxicity against different cancer cell lines. The major issue in the paper is not its scientific content, which is rigorous and detailed: the english level is poor, then the first strong step towards the eventual publication should be a deep revision of the grammar and construction of the phrases by a native english speaker which will help to make the manuscript easier to read, and the concepts easier to grasp.

Scientifically speaking the synthetic and complete characterization of these large number of derivatives is detailed (NMR, ESI mass, melting point, Rf) and all the biological part (inhibition and cytotoxicity tests) is very well done and it shows some interesting results.

One advise could be that the authors should report in the biological test tables also some literature values for similar adamantane based molecules to present a strong comparison with members of the same family.

In addition considering the large number of derivatives, maybe the author can do a better selection of the best results to be solely highlighted in the main text among the compounds presented

Here I report some example of bad english choices in the sole abstract (there are many to report in the main text)

In the abstract:

“The novel synthesized compounds were evaluated for activities against …” I do not think is correct, it should be “The novel synthesized compounds were evaluated against …”

“Compounds 2c, 2d, 2g, 2j and 3a, 3e, 3g possessed so well in inhibition …” possessed? Maybe behaved?

“Compounds 2a, 2e, 2h, 2k and 3j possibly have inhibition …” possibly? Maybe possessed moderate inhibition capability

“Compounds 2d and 2h, which contain –OH (octo) group …” Hydroxyl group?

Author Response

Respond to Reviewer 3
First, thank you for all your constructive comments. Your valuable
suggestions are the best point for helping us to complete our research.
From your comment, we respond and explain as the following:

Comment 1: The manuscript concerned the synthesis and biological
evaluation of adamantane derivatives, functionalized by thiosemicarbazone
moiety with differently substituted phenyl aldehydes or methyl phenyl ketones as
a screening for their biological activity against different bacteria and cytotoxicity
against different cancer cell lines. The major issue in the paper is not its scientific
content, which is rigorous and detailed: the English level is poor, then the first
strong step towards the eventual publication should be a deep revision of the
grammar and construction of the phrases by a native English speaker which will
help to make the manuscript easier to read, and the concepts easier to grasp.

Respond 1: Thank you for your consideration to our manuscript. We have
already proofreading and ask an English expert to check it. This is the final
version of our manuscript.

Comment 2: One advice could be that the authors should report in the
biological test tables also some literature values for similar adamantane based
molecules to present a strong comparison with members of the same family.

Respond 2: Thank you for your review. As actually, if using an
adamantane derivative like amantadine or rimantadine, our study would be more
convincingly. However, our lab had consumed all these chemicals when we run
the experiment in this manuscript, so we could not use these derivatives as
positive samples. It is the limit in this study. We hope to solve this matter in our
further research in the future.

Comment 3: In addition considering the large number of derivatives,
maybe the author can do a better selection of the best results to be solely
highlighted in the main text among the compounds presented.

Respond 3: Thank you for your point. As the actual, Reviewer's suggestion
is one the best point for our manuscript would be improved. Nevertheless, the
result in this manuscript is only the initial research of our approach. We are going
to study more deeply about adamantane derivative as well as the relationship of
bioactivity – structure of them. We hope to solve the point as you noted in our
deeply study in the future.

Comment 4: Here I report some example of bad English choices in the
sole abstract (there are many to report in the main text)
In the abstract:
“The novel synthesized compounds were evaluated for activities against
…” I do not think is correct, it should be “The novel synthesized compounds were
evaluated against …”
“Compounds 2c, 2d, 2g, 2j and 3a, 3e, 3g possessed so well in inhibition
…” possessed? Maybe behaved?
“Compounds 2a, 2e, 2h, 2k and 3j possibly have inhibition …” possibly?
Maybe possessed moderate inhibition capability
“Compounds 2d and 2h, which contain –OH (octo) group …” Hydroxyl
group?

Respond 4: Thank you for your constructive point. We have already
amended our manuscript, especially, the mistake in English language. The final
version was also checked by an English expert. I hope our manuscript is suitable
for Molecules Journal now.

In conclusion, we would like to thank for your attention to review our
manuscript. Your suggestion is a good point for our research in the future.
Thank you very much.

Round 2

Reviewer 1 Report

Although the introduction part was not expanded according to my suggestions and despite more improvements to the English language that could be done, I would recommend this manuscript for publication due to its scientific merit.